

# Mining known attack patterns from security-related events

Nicandro Scarabeo[1,2], Benjamin C.M. Fung[3] and Rashid H. Khokhar[4]

[1] Department of Electrical and Information Engineering, University of Cassino and Southern Lazio, Cassino, Italy
[2] Above Security, Inc., Blainville, Quebec, Canada
[3] School of Information Studies, McGill University, Montreal, QC, Canada
[4] Concordia Institute for Information Systems Engineering (CIISE), Concordia University, Montreal, QC, Canada

## ABSTRACT

Managed Security Services (MSS) have become an essential asset for companies to have in order to protect their infrastructure from hacking attempts such as unauthorized behaviour, denial of service (DoS), malware propagation, and anomalies. A proliferation of attacks has determined the need for installing more network probes and collecting more security-related events in order to assure the best coverage, necessary for generating incident responses. The increase in volume of data to analyse has created a demand for specific tools that automatically correlate events and gather them in pre-defined scenarios of attacks. Motivated by Above Security, a specialized company in the sector, and by National Research Council Canada (NRC), we propose a new data mining system that employs text mining techniques to dynamically relate security-related events in order to reduce analysis time, increase the quality of the reports, and automatically build correlated scenarios.

## INTRODUCTION

Security Operations Centers (SOC) represent a cornerstone of contemporary security services and are at the core of Managed Security Services (MSS). MSS are generally based on a set of distributed sensors that are deployed on clients' networks. Each sensor contains various security tools such as Intrusion Detection Systems (IDS), asset detection tools, flow analysis tools, etc. The said sensor tools analyze network traffic and send events to a central database (DB) repository for storage or further analysis. Security analysts can display information about security or network events (alerts or logs) and access their details through a Graphical User Interface (GUI) of the SOC. In order to improve overall attack coverage and satisfy the demand for more advanced service features, the volume of data that is collected in the sensor sensibly increases, and it becomes difficult for security analysts to maintain monitoring and analyzing increasingly large quantities of data without incurring *Service Level Agreement* (SLA) violations.

Many companies have tried to solve this issue by introducing pre-defined correlation rules in order to identify known network issues and to address client requests. This

Corresponding author
Nicandro Scarabeo,
nicandro.scarabeo@unicas.it

methodology represents a valid solution to maintain a high service quality, but it does not easily support the integration of new data sources. This method requires considerable maintenance and tends to exclude all collected events that are not part of any correlation rules. In fact, security analysts tend to dedicate most of their time analyzing pre-defined scenarios and writing reports without being able to leverage the unmapped data in order to identify traces of threats.

An alternative to tackle the problem is the introduction of data mining techniques. In fact the process which studies how to mine implicit and unknown information from very large volume of data, by using knowledge discover techniques, is known in literature as Data Mining. In the last decade few works have already applied data mining to cyber security in order to detect abnormal patterns in intrusion detection processes (*Thuraisingham, 2008*; *Chandola et al., 2006*). These papers were mostly deriving patterns by analyzing information derived by OSI model layer 3, OSI model layer 4, timestamps or, in general, structured protocol-related data stored in databases.

The objective of this paper is to develop an application that enables security analysts to efficiently analyze large amounts of security information by automatically mapping the message contained in the security-related events to *Common Attack Pattern Enumeration and Classification* (*CAPEC*) (*The MITRE Corporation, 2015*), which is a publicly available comprehensive dictionary and classification taxonomy of known attack patterns that can be used by analysts, developers, testers, and educators to advance community understanding and enhance defenses. Instead of identifying sequences of events or relating them only through common protocol-related information (i.e., common IP, port, etc.), the proposed system aims to gather events by relating them to known attack patterns, link their phases by analyzing the terms used to describe specific security issues, and applying text mining, which is the process of deriving knowledge (patterns and trends) from unstructured data. The application is developed in Python.

An attack pattern gathers the steps, the challenges, and the techniques that need to be executed in order to exploit a vulnerable system. CAPEC (*The MITRE Corporation, 2015*) represents an attack pattern as an entity represented by a global description of the attack; a detailed list of macro-steps, subdivided in actions that the attacker could follow; possible consequences of the attack; and methodologies for mitigating it.

In this paper, the semantic relatedness between sensor event messages and attack pattern fields description has been calculated in order to identify in real time which captured event is useful to fully describe the current attack. Thus, the goal of the present work is to identify the top-$K$ similar security events which can represent a specific attack step, defined in CAPEC. The set of events is presented to the analysts in the form of a recommendation. The analysts can then review it and automatically build a report, which is going to be stored for future analysis and recommendations.

*Contribution*. The major challenge is that there is no direct mapping between CAPEC attack patterns and security event logs. It is a non-trivial task to dynamically link a collection of unrelated security events to the attack pattern(s) in CAPEC because both entities are described in unstructured text. In this paper, we present a data mining system

that employs text mining techniques to dynamically relate the information between the security events and CAPEC attack patterns. To the best of our knowledge, this is the first work to automatically and semantically build attack scenarios by referring to a publicly available collection of attack patterns, resulting in the following capabilities:

- With an automatic system gathering collected events, analysts do not need to dig into a high volume of data in order to look for specific information. This significantly reduces the analysis time, referred as the time the analyst spends to look for stored events which could provide strong evidence of the attack under investigation. Thanks to the system here presented, the analyst can receive a list of automatically recommended events that are relevant to the attack under investigation.

- The system considerably increases the number of correlated and analyzed events included in an attack report prepared by security analysts for the clients. Furthermore, the quality and accuracy of the reports have been greatly improved. The mapped CAPEC attack patterns not only associate to security events as concrete evidence, but also make recommendations for how to address the security in the "Solutions and Mitigations" section of the associated CAPEC attack patterns.

- Extensive experimental results over different types of real-life security events illustrate the effectiveness of our proposed data mining system. We also demonstrate that our method can successfully build correlated scenarios.

The rest of the paper is organized as follows. 'Background' reviews background knowledge about the context in question. 'Related Work' reports related work. 'Problem Description' formally defines the research problem. 'Proposed Solution' describes the proposed solution. 'Discussion' discusses the possible limitation of the proposed method and future improvements. Comprehensive experimental results are presented in 'Experimental Evaluation'. Finally, 'Conclusions' concludes the paper.

## BACKGROUND

This section first introduces the monitoring analysis process which is executed in Security Operation Centers (SOCs). The section continues by articulating the motivation that brought the authors to develop such work. At the end, an example clarifies the usage of this application.

*SOC Workflow*. Figure 1 depicts the workflow in a SOC.

The process consists of four different phases, described as follows:

- **Collection Layer**. The collection layer captures the security-related events that are triggered by specific software. Typical events include (1) network logs, which have been sent by different hosts, and servers to aggregate them in a single location, and (2) alerts, which have been triggered by specific software that employ various real-time traffic monitoring algorithms. Those software can be signature based (e.g., Intrusion Detection Systems (IDS), Asset Detection, etc.) or anomaly based (e.g., Network Behaviour Analysis (NBA), Flow-based Classifier, etc.).

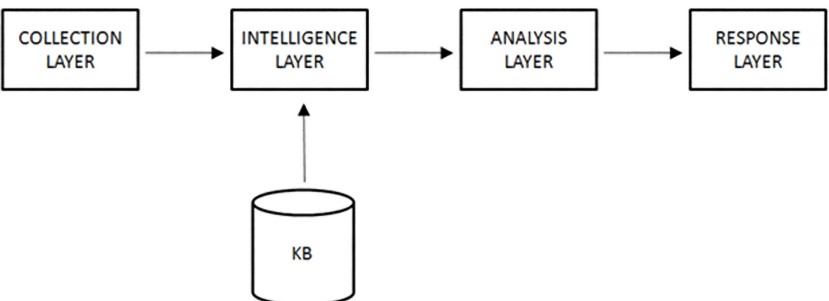

**Figure 1** **Workflow in a security operation center (SOC).**

- **Intelligence Layer**. The intelligence layer syntactically normalizes and stores the events for further analysis. The events are then compared to existing described scenarios that have been previously stored in the knowledge base (KB) and processed by statistical models, geared at recognizing anomalies. In case of positive matching, correlated events are sent to the analysis layer.

- **Analysis Layer**. Security analysts can observe the filtered data through the analysis layer. The analysts generally have tools for grouping data by specific network-protocol variables (e.g., IP, port, etc.) and they seek specific information in order to demonstrate the existence of an attack.

- **Response Layer.** In the response layer, the analysts contact the clients by writing reports or sending emails if they are able to present evidence of an attack, or to ask for further information.

- **Knowledge base (KB).** The knowledge base stores the pre-defined attack patterns. The intelligence layer checks the real-time data against the pre-defined attack scenarios in the KB in order to provide meta-events to security analysts.

*Motivation*. Network security monitoring has become an exceedingly essential service, and companies rely on the expertise of reputable Managed Security Service Providers (MSSPs) for guidance and best-practice recommendations in order to detect and prevent malicious activities. Monitoring is conducted by a team of security analysts who evaluate each problem almost manually, while basing their work on their expertise and previous knowledge about IT security requirements. Whether clients request more events be collected or whether they need to install more services to improve attack coverage, this approach will not scale well because the volume of data to be analyzed increases excessively. Hence, network security companies would like to overcome this problem by building pre-defined correlation rules, more or less complex, in order to identify a finite number of malicious or anomalous situations.

Although this approach scales well in the analysis layer, it has shown multiple limitations for maintenance and attack coverage supervision. Given the highly dynamic security context, this approach needs continuous improvement of existing correlation patterns by tuning parameters that should follow the evolution of the system. Moreover, new correlation patterns need to be introduced in order to discover new threats. Even

when using highly advanced correlation patterns, many collected events result in being discarded by these positive filters and never appear on the analysts' screens for review. This methodology is not necessarily wrong at first glance, since many events cannot be fully analyzed if they are not placed in the right context. However, discarding them cannot be the best solution either. Indeed, without rapidly creating new correlated patterns, even critical events that were collected after being triggered by the latest installed advanced security applicative might be discarded because they are not part of any correlation pattern. The idea of this paper is to automatically map all collected events by using a published structure existing in literature that collects all attack patterns (*The MITRE Corporation, 2015*).

***Example***. Suppose a security analyst needs to investigate the following IDS alert:

- ```
  IDS alert tcp $EXTERNAL_NET any ->  192.168.20.13:443
          (msg:"FILE-OTHER XML exponential entity expansion
             attack attempt"; ... sid:29800; rev:1;)
  ```

Using our presented methodology, the IDS alert results in a link to the following CAPEC attack pattern, which in turn is linked with other events that might have been collected in the sensor [2]:

```
CAPEC-197: XML Entity Expansion

Description: An attacker submits an XML document to a target
application where the XML document uses nested entity expansion to
produce an excessively large output XML. XML allows the definition
of macro-like structures that can be used to simplify the creation
of complex structures. However, this capability can be abused to
create excessive demands on a processor's CPU and memory. A small
number of nested expansions can result in an exponential growth in
demands on memory.

Outcomes: The attacker causes the target application denial of
service.
```

This attack pattern, defined by CAPEC, can be related to the following event instances:

```
a.  End Point Solution   Alert – High Usage of CPU on Host:$HOME_NET
b.  Network Behaviour Analysis Alert - Possible Denial of Service
    in the Network $NET_NAME
c. IDS alert tcp $EXTERNAL_NET any ->  $HOME_NET 80 (msg:"SERVER
   APACHE Apache mod_cache denial of service attempt"; ...;
   sid:12591; rev:7;)
```

Thus, through the use of our proposed methodology, the security analyst will know that the IDS alert, identifiable by the `sid` = 29, 800, can be associated to the aforementioned event instances, which are suggested to be checked by the analyst. This means that the concurrent arrival of those events, mapped to a known attack pattern, can provide strong evidence of the attack under investigation. ∎

# RELATED WORK

This section first discusses the state-of-the-art techniques in the area of security-related events correlation, followed by a discussion on some related works that concern applying text mining methods in the context of network security and use of recommender system.

*Event correlation technique in network security*

***Rule-Based Reasoning (RBR).*** A common approach to the problem of event correlation is to represent knowledge and expertise in a rule-based reasoning system (*Lewis, 1999*). RBR systems are often used in artificial intelligence applications such as the expert system, production system, or blackboard system. The system basically makes the inference between the collected data and the Knowledge Level, by a control layer that verifies a list of rules or a rule repository representing domain-specific expert knowledge. The knowledge in a RBR system can be updated without changing the program code of the inference engine, but it has several associated weaknesses. First, it is time consuming to manually enter the knowledge into the knowledge base. Second, the rule repository requires constant maintenance, which actually undermines the original purpose of a correlation engine to reduce the workload of administrators. The method proposed in this paper does not need human intervention to create new rules that represent attack patterns. The system will automatically map events to existing description of single phases of the attack. Moreover, the mapping will be automatically updated when a new event is collected or a new attack pattern is created.

***Case-Based Reasoning (CBR).*** The general idea of this approach is to solve new problems based on the solutions of similar past problems. CBR is a well-known research field in artificial intelligence that involves the investigation of theoretical foundations, system development, and practical application in the building of experience-based problem solving (*Bergmann, 2003*; *Hüllermeier, 2007*). CBR allows users to reuse knowledge from past cases in order to reduce acquisition efforts, which in turn increases the user's confidence in the system. However, manual knowledge engineering is required for adapting rules, the process is difficult to automate. Thus, it is still a time-consuming task. In contrast, the method proposed in this paper does not need human intervention to update existing attack patterns because the system will constantly update the mapping when a new event/attack pattern is created or modified. Moreover, by referring to external publicly available sources, the presented methodology can also map events to scenarios that have never been recorded in the monitored network.

***Graph-Based Approach (GBA).*** *Gruschke (1998)* presented a working principle for event correlation with a dependency graph, a directed graph that models dependencies between the managed objects. Its nodes (objects) reflect the managed objects of the system. Its edges reflect the functional dependencies between managed objects. Although it can manage a complex system and handle dynamic dependencies, the GBA is only capable of handling one problem at a time, i.e., GBA can identify only one root-cause. Identification of all root-causes may not be possible if multiple problems occur within the time period. Moreover, management requires a high level of expertise from the analysts. Due to these limitations, GBA is hardly useable in practical contexts.

*Text mining in security-related application*

Text mining has been applied to network security, but its research problems differ from the ones presented in this paper. By using text categorisation it is possible to distinguish and learn the characteristics of normal and malicious behaviour in regards to the words used in specific contexts. In *García Adevaa & Pikatza Atxab (2007)*, for example, the methodology is applied to log entries generated by web application servers in order to detect abnormal behaviour or fraud (*Hand & Weston, 2008*). In *De Vel, Anderson & Corney (2001)*, an investigation into e-mail content was performed for the purpose of forensic investigation. With such methodology, it is not only possible to recognize the topic discussed in the e-mail, but also abnormal conversation that might be related to targeted keywords. Similar goals have been studied in order to recognize maliciousness in social networks involving a large-scale data stream (*Keim, Krstaji & Bak, 2010*). In *Thompson (2005)*, through text mining it was possible to link entities across documents in order to recognize when different name strings are potential references to the same entity. In *Gegick, Rotella & Xie (2010)*, a system was implemented with an approach that applied text mining on natural-language descriptions of bug reports (BRs) in order to train a statistical model on already manually-labeled BRs to identify security bug reports (SBRs) that were manually mislabelled as not-security bug reports (NSBRs). In contrast, the research problem studied in this paper is to dynamically link a collection of unrelated security events to abstraction identifiable in the attack pattern(s) by using text mining techniques that can deal with unstructured text.

*Recommender system in security-related application*

Recommender systems have been an important research field in connection with information retrieval and information filtering systems for two decades. These systems are commonly applied in the industry of electronic commerce to facilitate customers in making decision on online purchases. Recommender systems are also widely applied to non-commercial applications for predicting and recommending items or actions based on user preferences and rating. There are few examples in literature which use recommender systems to help IT-professionals to deal with security and performance issues. For example, *Sampaio (2014)* developed a recommender system to solve performance issues occurring to network applications. The process proceeds with the identification of performance-related problems in networks and the characterization of user-profiles. Based on specific attributes which are used to characterize the sample cases, the work introduces the calculus of distance between a new case and stored ones. Another example is given by *Lyons (2014)* which elaborated a hybrid recommender system to generate list of cyber defense actions to mitigate cyber-security attacks. *Lyons (2014)* has a similar goal of this paper but it is specifically designed for defender counteractions and it takes into the consideration of only one attack-scenario. The recommended actions are based on historical data and are elaborated by a collaborative filtering approach.

## PROBLEM DESCRIPTION

In this section we first formally define the notions of security event and attack pattern followed by a description of the research problem. A *security event* contains a set of

attributes and a text description. For example, an attribute can be an IP source, an IP destination, or a port source, etc. (*Yurttas*). A text description can be "*MS-SQL raise error possible buffer overflow*".

**Definition 1. (security event):** Let $E(A_1, \dots, A_m, D)$ be a collection of security events, where $A_i$ is an security event attribute and $D$ is set of distinct words in all text descriptions in $E$. A security event $e \in E$ contains $\{a_1, \dots, a_m, d$, where $a_i \in A_i$ and $d \subseteq D$. While $d$ represents the description of the event, the other $a_i$ refers to protocol-related attributes.

An attack pattern in CAPEC, the publicly available comprehensive dictionary and classification taxonomy of known attack patterns, contains a set of attack steps, where each step is a text description that describes the details of the attack step, consequence, attack prerequisites, etc.

**Definition 2. (attack pattern):** Let $P$ be the set of attack patterns. Each attack pattern $p \in P$ contains a set of attack steps, denoted by $p = \{s_1, \dots, s_u$, where each attack step $s_j$ contains a set of words.

To provide a concise description, in the rest of the paper, we assume that the text description of the security events in $E$ and the text description of the attack steps in $P$ share the same set of vocabularies $D$. Alternatively, the method may define a map to translate between the two sets of vocabularies. The following distance function measures the distance between an attack event and an attack step.

**Definition 3. (distance function):** Let dist$(e.s)$ be a distance function that measures the distance between an attack event $e$ and an attack step $s$.

The attack event $e$ and the attack step $s$ are represented as two sets of words. The distance function, which will be further instantiated in 'Documents similarity function", calculates the distance between two sets of words.

The research problem is to identify the topmost similar security events for each attack step in the attack patterns defined in CAPEC. The similarity is defined as the inverse of the distance function, sim$(e.s) = 1/\text{dist}(e.s)$. In short, if two sets of words have more words in common, the smaller the distance. If the sets do not share words, the distance is the maximum. Definition 4 provides a formal description of the research problem.

**Definition 4. (attack patterns and security events matching problem):** Given a collection of security events $E$ and a set of attack patterns $P$, the research problem is to identify the top-$K$ similar security events $\{\hat{e}_1, \dots, \hat{e}_K\} \subseteq E$ for each attack step $s$ of each attack pattern $p \in P$, such that for $\forall \hat{e}_x \in \{\hat{e}_1, \dots, \hat{e}_K\} \forall e_y \notin \{\hat{e}_1, \dots, \hat{e}_K\}$, dist$(\hat{e}_x, p.s) \leqslant \text{dist}(e_y, p.s)$, where $K$ is a user-defined positive integer threshold.

An attack step is an abstract description in CAPEC. The top-$K$ events $\{\hat{e}_1, \dots, \hat{e}_K\}$ provide a more concrete description of the security information in the context of a security event. By observing the top-$K$ security events of each attack step in a given attack pattern, a

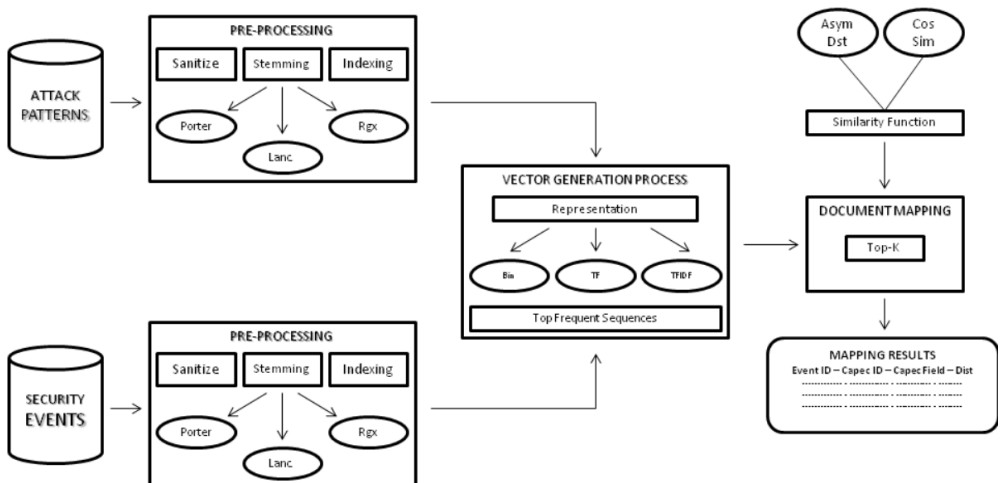

**Figure 2** Pipeline diagram illustrating the overall process of the proposed data mining system.

security analyst can identify and describe the evolution of an attack through the evidence of collected events.

## PROPOSED SOLUTION

Figure 2 depicts an overview of the proposed solution, followed by a detailed description of each component.

As shown in Fig. 2, through an off-line process the set that includes all the description $d \in E$ goes through pre-processing steps that sanitize sensitive data before it is stemmed. The stemming process is achieved by using optionally the Porter stemming algorithm (*Porter, 1980*), the Lancaster stemming algorithm (*Paice, 1990*), or the Regex-driven algorithm (*Yu et al., 2015*), as described in detail later on. From the stemmed data, specific lists of stop words are filtered out so it can go through the tokenization process.

Once tokenized, from the description field a vector $v$ is extracted that maps the content of $d$ in a dictionary $D$. To enforce the mapping and give more relevance to specific security-related expressions, the maximal frequent sequences are also extracted, in accordance with *Ahonen-Myka (1999)*, and considered as a single term of the dictionary. As representation of the vector $v$, the framework allows the user to choose among a binary, TF, or TFIDF representation.

Each of these can be set up in the configurable framework. As seen from Fig. 2, each $s_j$ characterizing $p = \{s_1, \ldots, s_u \in P\}$ goes through the same process, which transforms $s_j$ in a specified vector $w$. Once the vectors are built it is necessary to set up the distance function $(e, s)$, which can be either the Asymmetric Distance (*Wikipedia Contributors*) or the Cosine Similarity function (*Salton & McGil, 1986*).

Both functions interpret valuable alternatives to represent the $\text{dist}(v, w)$ that is going to be used in the 'Document Mapping Step' to find the top-$K$ similar security events that satisfy the problem statement demarcated in Definition 4. The algorithm that has been chosen to achieve this result is the k-nearest-neighbour classification (*Cover & Hart, 1967*).

## Preprocessing

The following preprocessing procedures are applied on the document $d \subseteq D$.

### General sanitizing

In each document, the terms, containing symbols or number, are removed. This step, for example, removes terms that represent byte signatures, such as '*50 4B 03 04*', or binary flag, such as 0x0014.

### Stemming

In each document, words that have morphological forms are normalized to their canonical form. Three algorithms were analyzed and can be set up in the code:

- The Porter stemming algorithm, which removes the commoner morphological and inflexional endings from words in English for this purpose (*Porter, 1980*).
- The Lancaster stemming algorithm (*Paice, 1990*), an aggressive stemming algorithm that reduces English words to their roots. For example, the word 'maximal' becomes 'maxim', and the word 'cement' becomes 'cem'.
- A Regex-driven stemming algorithm (*Yu et al., 2015*) that uses regular expressions to identify morphological affixes. Any substrings that match the regular expressions will be removed.

### Stopwords removal

Common English words, such as "a" and "the", are removed. We compiled an additional static list of stopwords that are common in the security contexts, e.g., "security", "attacker", to better index the documents.

### Duplicates stripper

Identical documents are removed. This can happen because different revisions of the same event might exist in the system.

### Tokenization

Given a sequence of terms in a document, the tokenization process separates the terms into tokens by using white spaces and punctuation marks as separators. This operation transforms the sequence of terms $d$ in vector of terms $\vec{d}$.

## Mining maximal frequent sequences

In the documents, it is possible to find interesting sequences, e.g., 'denial servic', 'stack overflow attempt', login attempt', 'download attempt', which represent well the threats described in the event. In order to give more weight to these terms in the indexing phase, a maximal frequent sequence mining algorithm has been implemented. The algorithm chosen for this purpose has been coded following the Apriori-like method described in *Ahonen-Myka (1999)*, and the set of sequences MFS $= \{q_1, \ldots, q_{|\text{MFS}|}$ has been determined. In the implementation, we employed an Apriori-like method. However, the main contribution of this paper is to propose a methodology to tackle the problem described in the paper. The one may replace specific algorithms, e.g., the maximal frequent

sequence mining method, by other more efficient algorithms based on their preferences and needs. The result will be the same.

## Document representation

After the preprocessing, the set of distinct and ordered terms in a given document set is denoted by Dict = $\{w_1, \ldots, w_{N_d}\}$. Next, all documents are indexed by their terms. Three representations have been implemented and they are described in the next sections.

### Binary representation

For each vector $\vec{d}$, we build a binary vector of |Dict| elements denominated $\vec{b} = [v_1, \ldots, v_{|\text{Dict}|}]$ where

$$v_j = \begin{cases} 1 & \text{if } w_j \in d \\ 0 & \text{otherwise} \end{cases}$$

### TF representation

For each vector $\vec{d}$, we build a numerical vector of |Dict| elements denominated

$$\vec{t} = [v_1, \ldots, v_{|\text{Dict}|}] \tag{1}$$

such as

$$v_j = \text{count}(w_j, d)$$

where $\text{count}(x, y)$ is a function that counts the occurrences of the term $x$ in the set $y$. A dual representation related to TF is called **Inverse Document Frequency** and is calculated as follows:

$$v_j = count(w_j, d)/|\text{Dict}|.$$

Generally this representation is preferred when it is important to diminish the weight of terms that occur very frequently in the document set and increases the weight of terms that occur rarely (*Luhn, 1960*).

### TF-IDF representation

For each vector $\vec{d}$, we build a numerical vector of |Dict| elements denominated

$$\vec{g} = [v_1, \ldots, v_{|\text{Dict}|}] \tag{2}$$

such as

$$v_J = \frac{\text{count}(w_j, d)}{|\text{Dict}|} \cdot \log\left(\frac{|D|}{\text{count}(w_j, D)}\right)$$

where $\text{count}(w_j, D)$ counts the number of elements of $D$ that contain the term $w_j$. This representation, proposed in *Spärck Jones (1972)*, gives a higher weight to a high term frequency (in the given document) and a lower weight to document frequency of the term

in the whole collection of documents; the weights hence tend to filter out common terms (*Wikipedia Contributors*).

The representation chosen for the first experimental results is the binary representation, but all of them have been coded and can be set up by the user.

After implementing such models, it was noticed that it was necessary to enforce the representations of the documents by also taking into consideration the terms that often come in sequences. Thus, in order to give more weight to those cases, the representation, aforementioned, has been improved by counting the maximal frequent sequences as additional single terms of the dictionary. By doing so, the vector $\vec{b} = [v_1, \ldots, v_{|\text{Dict}|}]$ will have extra $|\text{MFS}|$ number of elements, resulting in:

$$\vec{b}^* = \left[v_1, \ldots, v_{|\text{Dict}|}, v_{|\text{Dict}|+1}, \ldots, v_{|\text{Dict}|+|\text{MFS}|}\right] \tag{3}$$

and consequently, if binary representation is used,

$$v_j = \begin{cases} 1 & \text{if} \begin{cases} j \leq |\text{Dict}| \text{ and } w_j \in d \\ j > |\text{Dict}| \text{ and } q_{j-|\text{Dict}|} \in d \end{cases} \\ 0 & \text{otherwise} \end{cases}$$

where $\text{MFS} = \{q_1, \ldots, q_{|\text{MFS}|}\}$ represents the ordered set of maximal frequent sequences.

### TF-IDF representation

For each vector $\vec{d}$, we build a numerical vector of $|\text{Dict}|$ elements denominated $\vec{g} = [v_1, \ldots, v_{|\text{Dict}|}]$ (2) and $\vec{b}^* = \left[v_1, \ldots, v_{|\text{Dict}|}, v_{|\text{Dict}|+1}, \ldots, v_{|\text{Dict}|+|\text{MFS}|}\right]$ (3).

## Documents similarity function

Having determined the way to represent terms and documents, we utilized the asymmetric distance and cosine similarity measure to measure the similarity between the vectors.

### Asymmetric distance

The simultaneous absence of many terms in the vectors suggests the most relevant results to be the discovery of having something in common, rather than sharing the lack of specific terms: the agreement of two 1's (a present–present match or a positive match) is more significant than the agreement of two 0's (an absent–absent match or a negative match). The asymmetric distance has been calculated in according to *Wikipedia Contributors* and its value spans between 0 and 1.

### Cosine similarity measure

The present distance has been preferred to other known distances because the cosine similarity measure abstracts the length of the documents and focuses on the correlation between their vectors. Since all axis coordinates are positive, the cosine similarity value between any two term vectors is always between zero and 1:

$$0 \leq \vartheta\left(\vec{g}_i, \vec{g}_j\right) \leq \pi/2 \rightarrow \cos\vartheta \in [0, 1]$$

where $\vartheta$ is the angle between $\vec{g}_i$ and $\vec{g}_j$ (*Salton & McGil, 1986*).

Since the binary representation was chosen for the experimental results, as a consequence the asymmetric distance was selected. However, both have been coded and can be set up by the user, and, in case of a different representation, the cosine similarity distance is automatically selected.

## Semantic relatedness

After choosing the representation model and the documents similarity function, we employed a machine learning algorithm, K-nearest-neighbour (*Cover & Hart, 1967*), to determine the closest events of a specific attack step. The algorithm is usually used for determining the class of a specific unclassified element by evaluating the classes of the nearest elements. In our case, this is not necessary, and the method is just used to pick the top-$K$ related events $\{\hat{e}_1, \ldots, \hat{e}_K\}$ of a specific attack step $s_i$. However, in such a system it can be verified that some attack steps do not share enough terms with any of the events being captured in the sensor. This can happen if there is no tool in the sensor that can trigger an event that describes an action defined in a specific attack field. The result of that would be in classifying as neighbours some events that are the closest compared to others, but in fact very far (dist($e.s$) $\approx$ 1). For this reason, a maximal tolerated distance is fixed, and all the neighbours above this threshold are discarded.

## DISCUSSION

As mentioned above, the proposed methodology automatically maps security-related events to pre-defined attack patterns. This means that security analysts have a much better understanding of the events being collected from a specific network because of their classification and the possibility to directly relate a raw event to a sequence of attack steps. The use of CAPEC is useful to help analysts in examining data and generating responses without obligating them to remember too many notions and saving time they would spend on the Internet to look for confirmation and recommendations.

Moreover, through this methodology the analyst can clearly see the attack steps that are not mapped to any collected events and the mapped events that were not received in the sensor. The first case can suggest to the analysts which specific technology the company is missing in order to detect that specific attack steps; the second case instead suggests the attack steps that were missed by the tool installed in the sensor.

Given the vastness and the diversity of the collected events, we can argue that the mapping process might take too much time. This is actually not relevant since the mapping process can be executed off-line because the instances of the whole set of events are available. In real-time only the occurrences of the mapped events are checked and shown to security analysts in conjunction with the attack pattern associated with them. The system does not require specific maintenance either because the discovery of new attack patterns and the creation of new collectable events have a slow evolution, which could be dealt with by running the algorithm once a day on the new data only.

By relating events to the closest attack step, it is possible that a specific event is not mapped to any attack step. In that case we suggest the following:

- Evaluate the criticality of the event, and its worthiness to be collected;
- Evaluate the usage of synonyms to describe the content of the event; and
- Evaluate the possibility of creating a new attack pattern in which the event could be inserted.

The last suggestion reveals that the presented methodology does not need to be applied on CAPEC documents only, but it can be extended to any kind of document that aims to describe a Security related topic.

# EXPERIMENTAL EVALUATION

The objective of the experiments is to illustrate the precision of the algorithm, the capability of generating new correlation patterns in an automatic fashion, and the diversification of the attacks being reconstructed through alerts. All experiments were conducted on a PC running Windows 7 on an Intel(R) Core (TM) i3 CPU with 4 GB RAM.

## Data description

For the following experiments, the events are represented by Snort Alerts; in particular, we have selected the top 1,000 most frequently analyzed alert instances in a MSS, extracted from *Sourcefire (2015)*. To represent attack patterns, we have used the CAPEC List Version 2.6 (http://capec.mitre.org/data/#downloads), including 450 different attack patterns with 2,917 fields ($n$) that describe the attack steps and related information.

Following the framework depicted in Fig. 2 we employed the regex-driven algorithm as the stemming algorithm, binary vectors as data representations, and asymmetric distance as the similarity function.

## Precision

The system takes two user-specified thresholds $k$ and $dist_{MAX}$ as inputs, where $k$ is the number of neighbours to be defined in the K-NN algorithm, and $dist_{MAX}$ is the maximum tolerated distance between events and attack steps.

Figure 3 depicts the total number of neighbours being found (a.k.a. close relationships between Snort Alerts and attack steps) by the K-NN algorithm with respect to $dist_{MAX}$, having fixed $k = 10$. We set $dist_{MAX}$ to a range of values between 0.55 and 0.8. We observe that the number of neighbours spans from 3 to 1,465 when the Maximum Tolerated Distance ($dist_{MAX}$) increases from 0.55 to 0.80. However, comparing that assignment to the ground truth, the precision of the algorithm decreases with the increasing of $dist_{MAX}$. In fact, Fig. 4 shows that the percentage of error ($e$) spans from 0% to 27%. We can argue that the best result consists of having a high number of neighbours with very limited number of errors, in order to not mislead the analysts by associating security-related events to wrong attack steps. Having analyzed Fig. 4, we can assume that the best results are observed when $dist_{MAX} = 0.70$. The local minimum of the Error function, for $dist_{MAX} = 0.70$, is determined by the fact that, within the range $dist_{MAX} \in [0.55–0.70]$, the number of wrong relationships is caused by few misleading words and it is much lower than the number of total relationships established by the algorithm. By increasing the

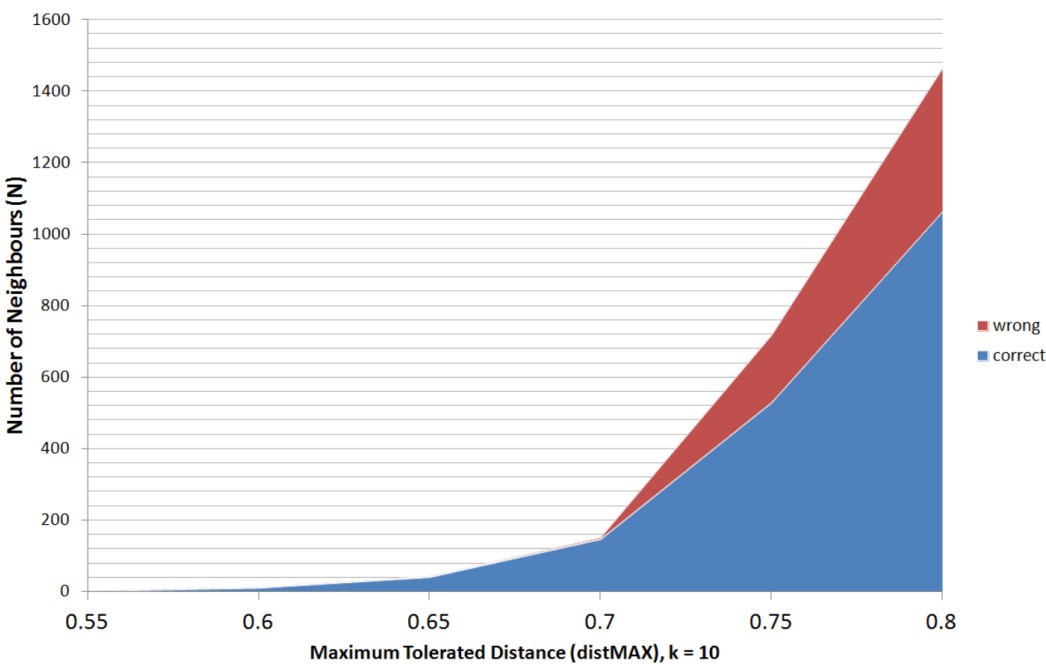

**Figure 3** Number of Neighbours (N) to Maximum Tolerated Distance ($dist_{MAX}$), and difference between correct and wrong relationships.

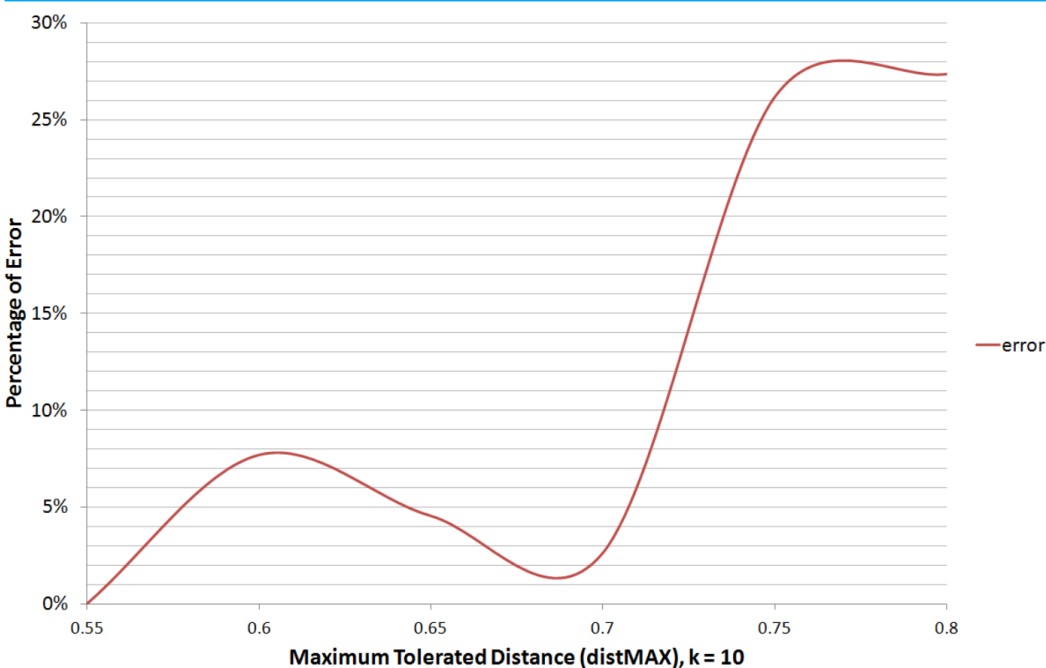

**Figure 4** Percentage of wrong relationships to Maximum Tolerated Distance ($dist_{MAX}$).

$dist_{MAX}$, the number of total neighbours increases sensibly, although limited on the upper part by $n * k$, but the error function assumes values that outdo the undesirable threshold of 25%; so we can assume that $dist_{MAX} = 0.70$ represents the best setting in order to have good precision and a high number of neighbours.

**Events Related to Attack Patterns**

**Figure 5  Number of different alerts being related to single attack patterns, represented by the CAPEC ID.**

## Diversification of neighbours

Having fixed $dist_{MAX} = 0.70$, which is the value that assures the best results according to Fig. 4, Fig. 5 depicts the total number of events being related to every single attack pattern, represented by a CAPEC ID. The result suggests that the proposed method dynamically defines multiple correlation patterns, each containing a certain number of alerts. As shown in Fig. 5, 53 attack patterns have been related to a number of different Snort alerts, which span from 1 to 27 attributions, based on the similarity between attack step fields and the events. Figure 6 depicts the distribution of the attack patterns according to the number of events related to them. We can argue that the relationship between an attack pattern and a single Snort alert (first bar of the graph shows in Fig. 6, in correspondence of $N = 1$) is not relevant because security analysts are only interested in relating events being triggered by the sensors. However, those relationships help analysts in retrieving useful information about the totality of the attack, e.g., missing steps or related information about the attack as prerequisite, mitigations, etc.

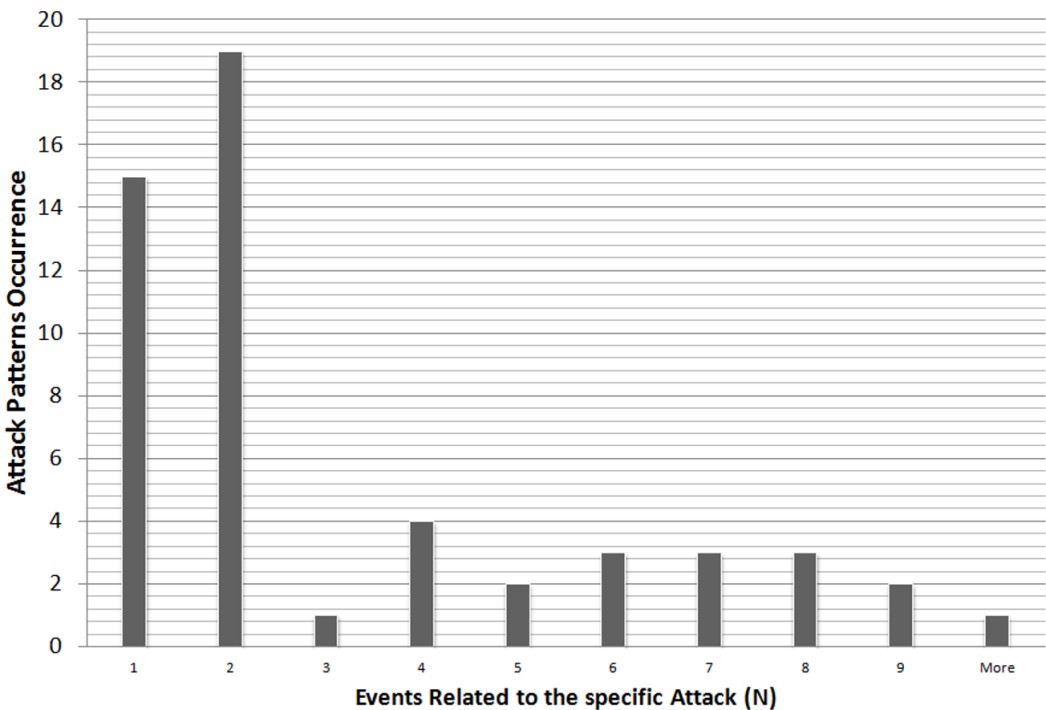

**Figure 6 Attack pattern occurrences to events being related to the attack.**

| CAPEC field | 2917 | 2917 | 2917 | 2917 | 2917 | 2917 | 2917 |
|---|---|---|---|---|---|---|---|
| SNORT alert messages | 1527 | 3054 | 6108 | 12216 | 18324 | 24432 | 30540 |
| | | | | | | | |
| Preprocessing | 01:33.586 | 02:06.753 | 03:11.068 | 05:19.088 | 07:22.241 | 09:26.917 | 11:35.045 |
| Dictionary + MFS | 03:48.584 | 04:06.285 | 05:21.457 | 09:59.710 | 15:19.017 | 23:59.598 | 36:51.010 |
| Vector Representation | 00:56.705 | 01:18.640 | 02:16.312 | 04:35.949 | 07:09.244 | 10:18.189 | 13:57.312 |
| Top k | 01:28.566 | 03:39.832 | 09:22.247 | 22:11.590 | 39:13.145 | 54:47.812 | 1:19:47.72 |
| | | | | | | | |
| Total Time | 07:47.441 | 11:11.510 | 20:11.084 | 42:06.337 | 1:09:03.65 | 1:38:32.52 | 2:22:11.09 |

**Figure 7 Computational Time calculated by varying the number of Snort Alerts Messages to be mapped to the whole set of Capec Fields.**

## Computational time

The computational time was analyzed by taking in consideration, incrementally, the whole set of distinct messages included in Snort Alerts (*Sourcefire, 2015*). For the analysis of the computational time, the four different phases of the framework have been clocked by varying the number of distinct Snort alert messages considered.

As shown in Fig. 7, the time necessary to map the whole set of distinct messages, extractable from Snort alerts, to the whole set of CAPEC fields, is quantifiable in about two hours and twenty minutes. By considering that the system needs to be updated when the CAPEC structure is modified (every three months, *The MITRE Corporation, 2015*) or

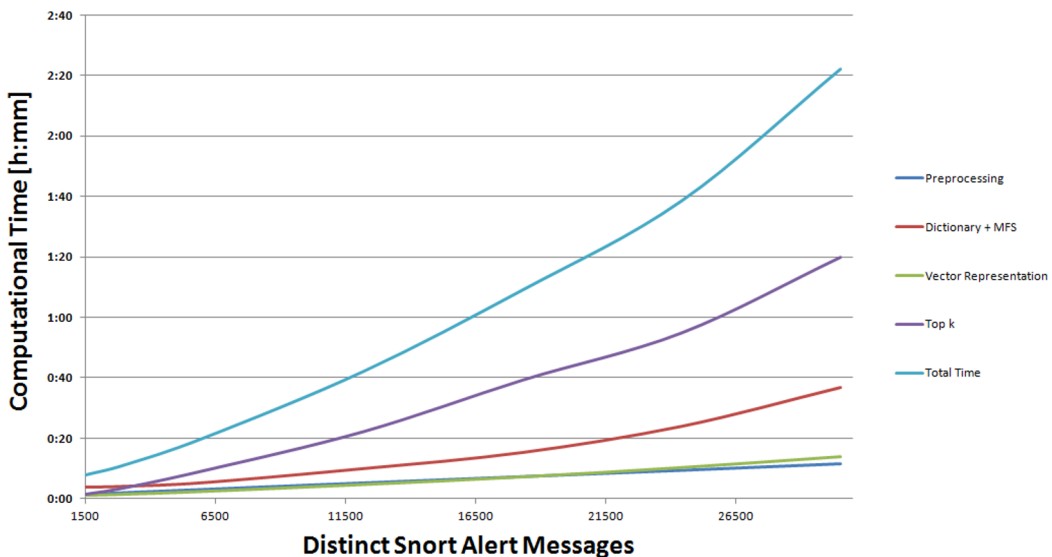

**Figure 8 Computational Time calculated by varying the number of Snort Alerts Messages to be mapped to the whole set of Capec Fields.**

when new Snort rules are published (every couple of day), the computational time can be considered very low, because it is below that margin. It is important to note that the whole process is executed off-line: when an analyst is called to interpret a specific alert in-line, the system will map it to pre-defined attack patterns by using previously stored results. Thus, the analyst can avoid manually mapping the alerts and efficiently obtaining the results. Figure 8 shows that the extraction of the top-*K* alerts, described in 'Semantic relatedness', it is the one which mostly contributes to the total computational time, and it must be improved in a new re-factoring of the system.

## CONCLUSIONS

In this paper, we have proposed and developed a data mining frameworks that employs text mining techniques to dynamically relate the information between the security-related events and CAPEC attack patterns, both described in unstructured text. Moreover, we introduced an automatic system to gather events that can significantly reduce the analysis time and increase the number of correlated and analyzed events included in an attack report prepared by security analysts for the clients, improving the quality and accuracy of the reports. For future work, it would be interesting to investigate the impact of including other standards in addition to CAPEC, e.g., *Extensible Configuration Checklist Description Format* (*XCCDF*) (*National Institute of Standards and Technology*). Also, it would be interesting to study the clusters of words that characterize the events that are not currently classified in order to understand the missing attack patterns or attack steps in CAPEC.

### Funding

Above Security has supported the paper in terms of human resources and provided us with simile-real-life materials for experimentation. The paper is also supported in part by the National Research Council Canada (NRC) Industrial Research Assistance Program, NSERC Discovery Grants (356065-2013), and Canada Research Chair Program (950-230623). The funders had no role in study design, data collection and analysis, decision to publish, or preparation of the manuscript.

### Grant Disclosures

The following grant information was disclosed by the authors:
National Research Council Canada (NRC).
Industrial Research Assistance Program.
NSERC Discovery Grants: 356065-2013.
Canada Research Chair Program: 950-230623.

### Competing Interests

Nicandro Scarabeo is an employee at Above Security, Inc. Benjamin C.M. Fung is an Academic Editor for PeerJ.

### Author Contributions

- Nicandro Scarabeo conceived and designed the experiments, performed the experiments, analyzed the data, contributed reagents/materials/analysis tools, wrote the paper, prepared figures and/or tables, performed the computation work, reviewed drafts of the paper.
- Benjamin C.M. Fung reviewed drafts of the paper.
- Rashid H. Khokhar worked on the first release of the 'Related Work' paragraph.

### Data Availability

Revealing sensitive data that might be used by Above Security to address security procedures is not included in our scope. Therefore, codes and raw data cannot be published in conjunction with this work.

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
