# Peer review of "Mining known attack patterns from security-related events"

_PeerJ Computer Science, doi:10.7717/peerj-cs.25_

## Round 0.1 · original submission · Major Revisions

Pay special attention to reviewer 1 comments, because he has serious concerns to accepts the paper. Try to remark the novelty of your approach, and to extend the experimental part of the paper to provide convincing results, with an appropriate statistical treatment of results.

·

Basic reporting

The article deals with a really interesting problem. Authors propose a new data mining system that employs text mining techniques to reduce the analysis time, increase the quality of the reports, and automatically build correlated scenarios.

I have some concerns about the introduction section. From my personal point of view, this is quite extensive. At the same time, it does not deal with some important points like Data Mining. Thus, it is my understanding that the whole introduction section should be rewritten, describing the problem, describing data mining, denoting how data mining could be useful for solving the problem, etc. As for Figure 1 and its description, it could be more appropriate for the background section. Introduction section should introduces the problem and how to solve it.

I do not really understand what [Mitre] means (line 38 and 104).

As for the related work section (and also the introduction section), it is quite large. Noted that the problem description does not start till page 8 (from a total of 21). All of this could cause a lack of clarity in what the goal is and how it is solved.

References need to be revised. Most of them are from 90s. The reader could also find some references from 1986, or even from 1960. A research work should deal with a current problem, so references 50 years ago cannot be included.

Definitions 2 and 3 should be improved. Starting from Definition 2, the "set of words" needs to be defined. Is it contained in D? According to Definition 3, the distance function requires a better description. Maybe an example could help since it is not clear how one could measure the distancia between and attack event and an attack step. Finally, the inverse of a distance should be defined similarly to "distance function" does.

Line 308, authors state that the stemming process is achieved by using optionally the Porter algorithm, the Lancaster algorithm and the Regex-driven algorithm. Nevertheless, no citations are available for these algorithms. The reader feels confused in this line and has to go back discovering that it was not defined previously.

For mining the maximal frequent sequences, the authors state that they have chosen an algorithm coded by following the Apriori-like method. At this point, I wonder whether this algorithm is the best option for a text mining problem. Apriori-like algorithms suffer from both computational and memory requirements. Since the goal if to deal with a hihg number of words in an efficient way, the reader needs to know more about it and how the authors have solved this issue. Besides, there is no execution time analysis within the manuscript.

Line 393, there is a "¿". The same occurs in line 395 and 396.

Line 396 and 397, theres is a "Error: Reference source not found"

Experimental design

The experimental design seems to be well described, but it lacks of clarity in some aspects. For instance, since the authors use an Apriori-like algorithm, the hardware used for the experimental study is not appropriate (to the best of my knowledge). From my background in Pattern Mining and Associaiton Rule Mining, an Intel Core i3 with 4GB RAM could be enough just for some specific datasets (small ones).

An analysis of the computational time is required, specially if the authors state that the aim of the proposed system is to "reduce the analysis time"

Datasets should be publicly available, or at least, they should be described well in terms of complexity.

Validity of the findings

No commets

Additional comments

The work is interesting but it requires to be significantly improved. I expected much more from the experimental study, which is shorter than the related work section.

Reviewer 2 ·

Basic reporting

The paper proposes a process to relate attack patterns with security events that are triggered by specific software.

The topic is interesting. However, it is necessary that the understanding the paper are improved:

- Authors should specify clearly the work carried out. Time and again, they say that “they propose a new data mining system”, what a new data mining system is? This is a very broad definition to define their work and it is used in abstract, introduction, conclusion, ....

- There are several comments that cause confusion. For example, authors comment that they use data mining and text mining, the main difference between both process is the structure of patterns. However, the process/steps are similar. Then, they comment that use KNN classification algorithm, but without determining the class, they only want to find the most related events according to one distance. In this case, why do they talk of classification method?. In general, authors describe different general areas, but their work is not limited in no of them. Finally, if the finality is to show the user the top-K related events, in this case, they could revise the area of recommend systems.

It would be necessary that authors restrict well the work that they carry out and focus the rest of paper in this area (related work, proposal, experimentation, ….).

Experimental design

In experimentation section, concretely section 6.2, authors comment that increasing the maximum distance, the number of neighbours increases sensibly. However, if the number of neighbours is delimited to 10, this restriction would select only the 10 most closeness and there have been no difference although the maximum distance was largest.

It would be convenient that authors concrete this information.

Validity of the findings

No comments

Additional comments

Authors should specify if they have developed an new application or they use a specific software. Really, there is not a novel proposal in the different steps of the process and there are a lot of software that it is possible to use. It would be convenient that authors should add information about this topic.

In section 4.3, both section 4.3.2 and 4.3.3 specify that it is used a binary vector, but in this case it would be numerical, would not it?.

It would be necessary to review the symbols in equations from line 391 to 397.

---

## Round 0.2 · accepted · Accept

At the production step, please follow the suggestions provide by reviewer 2, who suggests that you include several references

·

Basic reporting

In the new version of the paper, all of my previous comments have been considered carefully. I think the paper can be accepted

Experimental design

No Comments

Validity of the findings

No Comments

Additional comments

In the new version of the paper, all of my previous comments have been considered carefully. I think the paper can be accepted

Reviewer 2 ·

Basic reporting

In general, I am satisfied with the review of paper in many points. However, there is one point that it would be convenient to review.

Authors go on combining different concepts to classify its paper. For example: “new data mining system that employs text mining techniques”, this description is confused and authors combine both definitions in the paper. I recommend authors that they focus directly on Text Mining to describe their work.

I recommend the next references to authors. These references show the description of text mining, its relation with data mining and its main applications:

Gupta, V., & Lehal, G. S. (2009). A survey of text mining techniques and applications. Journal of emerging technologies in web intelligence, 1(1), 60-76.

Patel, F. N., & Soni, N. R. (2012). Text mining: A Brief survey. International Journal of Advanced Computer Research, 2(4), 243-248.

Experimental design

No Comments

Validity of the findings

No Comments